# A State-of-the-Art Review on Biowaste Derived Chitosan Biomaterials for Biosorption of Organic Dyes: Parameter Studies, Kinetics, Isotherms and Thermodynamics

**DOI:** 10.3390/polym13173009

**Published:** 2021-09-06

**Authors:** Yean Ling Pang, Jia Hui Tan, Steven Lim, Woon Chan Chong

**Affiliations:** 1Department of Chemical Engineering, Lee Kong Chian Faculty of Engineering and Science, Universiti Tunku Abdul Rahman, Bandar Sungai Long 43000, Malaysia; jiahui998@1utar.my (J.H.T.); stevenlim@utar.edu.my (S.L.); chongwchan@utar.edu.my (W.C.C.); 2Centre for Photonics and Advanced Materials Research, Universiti Tunku Abdul Rahman, Bandar Sungai Long 43000, Malaysia

**Keywords:** chitosan composites, parameter studies, kinetics, isotherms, thermodynamics

## Abstract

Chitosan is a second-most abundant biopolymer on earth after cellulose. Its unique properties have recently received particular attention from researchers to be used as a potential biosorbent for the removal of organic dyes. However, pure chitosan has some limitations that exhibit lower biosorption capacity, surface area and thermal stability than chitosan composites. The reinforcement materials used for the synthesis of chitosan composites were carbon-based materials, metal oxides and other biopolymers. This paper reviews the effects of several factors such as pH, biosorbent dosage, initial dye concentration, contact time and temperature when utilizing chitosan-based materials as biosorbent for removing of organic dyes from contaminated water. The behaviour of the biosorption process for various chitosan composites was compared and analysed through the kinetic models, isotherm models and thermodynamic parameters. The findings revealed that pseudo-second-order (PSO) and Langmuir isotherm models were best suited for describing most of the biosorption processes or organic dyes. This indicated that monolayer chemisorption of organic dyes occurred on the surface of chitosan composites. Most of the biosorption processes were endothermic, feasible and spontaneous at the low temperature range between 288 K and 320 K. Therefore, chitosan composites were proven to be a promising biosorbent for the removal of organic dyes.

## 1. Introduction

Water is an essential and valuable component for sustaining the life of all living organisms and developing a sustainable ecosystem. However, water pollution has now become one of the most critical challenges facing many countries in accordance with the growth of urbanisation and socio-economic development. In Malaysia, approximately 97% of the raw water supply for domestic, agricultural and industrial activities comes from rivers [1]. The increased number of slightly polluted and polluted rivers was mainly due to uncontrolled usage of various chemicals and improper discharge of effluents from the household and industrial sector into the river without adequate treatment [2]. Several studies have reported that the most common pollutants present in industrial wastewater are heavy metals, organic compounds and inorganic compounds. Textile dyes are among the major contributors to water pollution [3].

In the textile industry, dyes are organic compounds mainly used to provide colour and alter the crystal structure of the coloured substance [4]. However, the washing, dyeing, and finishing of textiles require high water consumption, which eventually generate a high discharge rate of wastewater containing organic dyes. Approximately 100–300 m^3^ of water are utilised to generate one tonne of textile [5]. The amount of dye in the effluent is closely linked to the fixation rates of the various dyes and fibres. Table 1 shows the degree of fixation and the percentage of loss in the effluent for various types of dyes. During the dyeing and finishing processes, no single dye has a 100% degree of fibre fixation [6]. As a result, a high number of dyes will end up in the water bodies, resulting in highly coloured effluent due to their incomplete degree of fixation and exhaustion. The high colour intensity of dyes prevents the light penetration into the river and reduces the dissolved oxygen, thereby severely affecting the aquatic diversity. The dyes can persist in the environment due to the high stability and low biodegradability of dyes. If long term consumption of drinking water contains organic dyes, they can bioaccumulate in the body and may induce cancer and tumours in humans and animals [7].

Therefore, the adsorption process is widely employed to remove non-biodegradable organic dyes [8]. Synthetic polymer is an effective adsorbent to remove dyes from wastewater. However, it is restricted by its costly supply chain, which can increase the capital cost of wastewater treatment. It is also non-biodegradable, which might lead to environmental problems related to its waste disposal [9]. In light of this, many researchers have focused on the evaluation of inexpensive and environmentally friendly alternative sorbents. Food wastes are renewable and inexpensive biopolymers that can be used for the production of biosorbents. According to the Food and Agriculture Organization of the United Nations, about 1.3 billion tonnes of food is lost or wasted annually [10]. Inefficient food waste management may contribute to some negative environmental issues such as infectious diseases, land and water pollution. Hence, the environmentally friendly approach is introduced to recycle these food wastes as secondary useful materials.

Food wastes are considered biopolymers that are rich in carbohydrates, lipids and proteins. The investigations on the recovery of chitosan from food waste especially marine food waste are deemed interesting due to the renewable attractive properties. According to the current research, the extraction of chitin and chitosan are commonly carried out by chemical or biological methods. The extraction process of chitin involves demineralization and deproteinization, followed by the deacetylation to convert chitin to chitosan by removal of acetyl group [11]. Knidri et al. [11] and Sivanesan et al. [12] had reviewed those extraction chitin and chitosan processes in details recently. Among the biopolymers, chitosan has been reported in the literature as a promising biosorbent for the removal of dyes [13,14]. It consists of carboxyl (–COOH), hydroxyl (–OH), amine (–NH_2_) and amide (–NHCOCH_3_) functional groups on its surface, which are responsible for the uptake of organic dyes through intermolecular interaction, hydrogen bonding and electrostatic attraction. However, pure chitosan has some limitations such as low adsorption capacity, low regenerative and poor mechanical strength, thereby limiting its potential applications. Hence, a chemical or physical modification of chitosan can be carried out by introducing the reinforcement material in the chitosan matrix to synthesise chitosan composite. In this study, various chitosan composites were studied to analyse their biosorption behaviours to remove organic dyes. The performance of chitosan composites for the biosorption of organic dyes were investigated through various parameter studies such as solution pH, biosorbent dosage, initial dye concentration, contact time and solution temperature. This paper also reviewed the research state of the art of biosorption isotherms, kinetics and thermodynamics.

## 2. Biopolymers

Biopolymers can be described as natural polymers synthesised by living organisms through naturally synthesised enzymes and catalysed chain growth polymerisation in their growth cycle of biological cells [15]. Many materials such as microorganisms, plants or trees that are commonly derived from biological sources can be described by the term "biopolymers". Materials chemically synthesised from biological raw materials such as starch, sugar, fat, resin, protein and vegetable oil can also be called biopolymers [16]. Biopolymers are mainly made up of a long chain of repeating monomer units that are covalently bonded to form larger molecules. There are three types of biopolymers: polysaccharides, polypeptides, and polynucleotides with the monomer units of sugars, amino acids and nucleotides, respectively.

Food wastes are also one of the primary sources of biopolymers. These food wastes are considered as a valuable renewable resource as they contain high-value components such as polysaccharides, proteins, and lipids. They can be converted into a variety of valuable products such as biochemical products and biofuels. Nowadays, biopolymers have been widely used in various sectors and have gradually replaced 30–90% of petro-chemical polymers [17]. The unique properties of biodegradability, biocompatibility and renewability of biopolymers have attracted worldwide attention to be used as petro-chemical polymers [18]. Among these biopolymers, polysaccharides such as chitosan, cellulose, and lignin have received particular attention for environmental application [19]. Polysaccharides are produced based on a renewable basis and represent the largest group of polymers currently produced in the world. In fact, more than 150 million tonnes of polysaccharides are produced annually, as compared to about 140 million tonnes of synthetic polymers [20]. Polysaccharides are polymers of amino sugars or glucose bonded by acetic bonds. It is well known that polysaccharides are biodegradable, available in abundance and have the ability to associate with various types of molecules through physical and chemical interactions [21].

Recent studies have explored the application of food waste derived biochemical components extracted from lignocellulosic substrates such as succinic acid and 2, 3-butanediol as precursors for the production of biopolymers [17,22]. Biopolymers also can be used as natural coagulants and flocculants in the wastewater treatment to remove non-biodegradable and highly stable water pollutants such as organic dyes, pharmaceutical compounds and heavy metal ions [23]. These biopolymers are also widely used as biosorbents in the wastewater treatment plant because of their particular structures and physicochemical characteristics. They do not produce toxic by-products during the chemical treatment process [24]. In addition, biopolymers consist of several functional groups in their chemical structure, such as –OH, –NH_2_, and –COOH groups in their polymer chain. Hence, biopolymers have high reactivity and excellent selectivity towards aromatic compounds and metal ions. They are able to interact with organic compounds through chemical or physical sorption. More specifically, adsorption on polysaccharide derivatives can be considered as a cost-effective method to separate and remove contaminants from the water bodies. It is also an environmentally friendly method to preserve the environment. Besides, the increasing number of publications on the adsorption of toxic compounds by these biopolymers indicate a recent concern in the development of new adsorbent polysaccharides-containing materials [25].

In this study, the biopolymers are classified, and the types of biopolymers are described in the following sections. The biopolymers are then modified and used as biosorbents based on their adsorption capacity.

### 2.1. Classification of Biopolymers

The classification of biopolymers can be made based on the type of monomers, biodegradability, raw materials and backbones, as shown in Figure 1 [26]. Firstly, biopolymers can be categorised into two main groups, which are biodegradable and non-biodegradable biopolymers. They are then alternatively categorised according to their origin, either being bio-based or non-bio-base biopolymers. Next, the biopolymer can also be classified based on their polymer backbone, including polyesters, polysaccharides, polycarbonates, polyamides and vinyl polymers. The three types of biopolymers can be further distinguished into polysaccharides, polypeptides and polynucleotides, depending on the type of monomers.

Besides, the biopolymers can be further categorised based on the types of raw materials and their biodegradability. There are three categories of biopolymers: biodegradable bio-based biopolymers, non-biodegradable bio-based biopolymers and biodegradable fossil-based biopolymers.

### 2.2. Types of Biopolymers

Among the biopolymers, cellulose and chitosan are the first and second most abundant biopolymers in the world that have been successfully employed for wastewater treatment. Cellulose is a structural polysaccharide made up of repeating β-D-glucose units arranged in a linear chain and linked by β(1→4) glycosidic bonds through the condensation process [27]. Figure 2a illustrates the chemical structure of cellulose. The glycosidic bonds between β-D-glucose units make the structure of cellulose highly crystalline and stronger [23]. Approximately 33% of the plants are made up of cellulose. Citrus peels, wheat straw, rice and woody parts of vegetables are the richest cellulose source [28].

Chitin and chitosan are two promising biopolymers derived mainly from the exoskeleton of invertebrate animals and the cell walls of fungi. Chitin is a natural polysaccharide consist of a linear chain of 2-acetamido-2-deoxy-D-glucose joined by β(1→4) glycosidic bonds. Chitin is commonly obtained from commercial and marine sources such as crab shells, lobster shells, shrimp shells, oysters, squids, crawfish, cuttlefish and fungi [29]. Chitin is dumped as a waste product from the seafood industry and approximately 1 × 10^13^ kg of chitin is produced annually. Moreover, approximately 8 × 10^4^ tonnes of chitin are synthesised based on marine by-products [30].

Chitosan is the second most abundant biopolymer that is produced from chitin through the alkaline deacetylation process. Chitin deacetylation is a process that involves the reaction of chitin with 40%–50% sodium hydroxide solution to hydrolyse –NHCOCH_3_ groups into –NH_2_ groups. However, the degree of deacetylation can only achieve up to 98% because complete deacetylation is difficult to achieve due to the heterogeneous process. Thus, chitosan is considered a partial deacetylated form of chitin [31]. Generally, chitosan is formed from the units of D-glucosamine and a small amount of N-acetyl-D-glucosamine residue that joined by β(1→4) glycosidic bond [32]. Each glucosamine unit consists of a free amino group, and this group can take on a positive charge which allows chitosan to be used as a coagulant agent and adsorbent for the removal of organic compounds. Its coagulation effect is more effective than mineral coagulants such as aluminium sulphate, polyethene imide and polyacrylamide in the removal of various pollutants from the aqueous solution [33].

Figure 2b,c show the respective chemical structures of chitin and chitosan [27]. The chemical structure of chitosan is similar to chitin and cellulose, except the present of –NH_2_ groups in the chitosan structure. Nevertheless, the solubility of chitosan in the aqueous acid medium is higher than its precursor polymer (chitin) and cellulose, owing to the presence of –NH_2_ groups and –OH groups that are responsible for the adsorption process. Chitin is not suitable when used as a biosorbent for the adsorption of organic dyes as compared to chitosan due to its low solubility and low dispersion in water and other solvents. On the contrary, chitosan is soluble in soluble in most aqueous acid solutions such as acetic, citric, formic and lactic acids that are below its pKa (pH = 6.5) [11]. In order to improve the adsorption capacity of chitin for the removal of methylene blue dyes, Cao et al. [34] modified the chitin using a chemical process through protonation, carboxylation and grafting.

Chitosan has a long polymer chain and good polycation, aggregation and precipitation properties under neutral or alkaline pH conditions. It is biodegradable, biocompatible, non-toxic and inexpensive. The respective properties can promote the contact between polymer and organic pollutants. Chitosan can be used as coagulants and flocculants to remove the negatively charged colloidal organic and inorganic pollutants from water due to its high cationic charge density [35]. With their regenerative ability and environmentally friendly properties, they are ideally used in the adsorption process. Many research studies have been carried out to investigate the adsorption capacity of chitosan and its composites to remove organic pollutants from wastewater. The application of chitosan in water treatment such as organic dyes removal, heavy metals removal and oil treatment of aqueous emulsion were investigated [14,29,36].

### 2.3. Modification of Chitosan

Among the biopolymers listed previously, chitosan was selected for further discussion in the following sections as it could be used as a potential biosorbent to remove organic dyes. However, pure chitosan was reported to have poor physicochemical properties in terms of morphological properties, surface area, functional groups and thermal stability, thereby reducing the biosorption capacity [37]. Therefore, chitosan was modified using chemical or physical methods to overcome its limitations and improve the performance of chitosan. There are several chitosan modification methods to improve the performance of chitosan on the adsorption efficiency, which are crosslinking, grafting and blending with other materials [38].

The grafting method is commonly used to promote the formation of chemical bonds between different types of polymers without mixing with other materials. This method can improve the physicochemical properties of chitosan by introducing additional functional groups such as –NH_2_, –NHCOCH_3_, –COOH or thiol into crosslinked chitosan [39]. Besides, polymer grafting also can improve the polymerisation degree of biopolymer, polydispersity of main chain and side chains, graft density, graft distribution or graft uniformity [40]. In general, polymer grafting can be prepared using three methods as shown in Figure 3 [41]. In details, the “grafting to” strategy involves attaching a pre-synthesised polymer onto the surface or backbone of the polymer. The “grafting from” strategy is based on the attachment of the initiator to a surface/polymer chain end followed by the growing of polymer chain. Finally, the “grafting through” strategy involves the polymerisation of a macromonomer.

Kyzas et al. [42] grafted chitosan with polyacrylamide and also with polyacrylic acid to improve the adsorption capacity of chitosan. Grafting of functional groups such as –COOH, –OH, sulphate, phosphate and –NH_2_ groups onto the biosorbent surface could increase the number of active sites of chitosan for the removal of organic dyes. For example, the surface of biosorbent was protonated by grafting –NHCOCH_3_ functional groups on the chitosan surface. As a result, this could improve the adsorption capacity of chitosan towards the negatively charged organic compounds. Besides, grafting of –COOH groups onto the surface of chitosan would result in more negative charges on the chitosan surface. Thus, this leads to the high adsorption capacity of chitosan towards cationic organic pollutants due to the strong interaction between cationic dyes and –COOH functional groups.

Moreover, crosslinking is one of the effective approaches to modify polymer microstructures via chemical or physical methods. However, excessive crosslinking will reduce the mechanical stability of polymers and therefore crosslinking must be performed in an appropriate and controlled manner [43]. Crosslinking reactions are commonly employed to prevent chitosan from dissolving in acidic media. At lower pH, chitosan will be dissolved in an acidic medium and lose its ability to bind with adsorbates, subsequently restricting its applications in wastewater treatment. Therefore, crosslinking reagents containing multifunctional groups such as aldehydes, anhydrides and epoxides will be used to react with the functional groups on the surface of chitosan [44]. The common crosslinkers used include glutaraldehyde, epichlorohydrin, glyoxal formaldehyde, ethylene glycol diglycidyl ether and sodium tripolyphosphate [45].

Next, polymer blending is an economical and simple physical method of combining two or more materials, with or without any chemical bonds between their chains to develop a new material called polymer composite. However, it is provided that the concentration of polymer component in the blends should be above 2%. Polymer blending with other materials is widely used to improve the mechanical properties of biopolymers due to their recyclability and biodegradability [46]. The components involved in the blending usually will have different physicochemical properties with each other. Therefore, blending also helps to overcome the shortage of reaction sites in the adsorption membrane structures of biopolymers. For example, Anitha, Kumar and Kumar [47] employed the polymer blending method to synthesise chitosan/polyvinyl alcohol (PVA) for the removal of Direct Red 80 dyes. Based on the results, chitosan/PVA could remove up to 89% of Direct Red 80 dyes, while pure chitosan only could remove 54% of dyes. The results indicated that blending PVA with chitosan could significantly improve the removal efficiency of dyes.

## 3. Chitosan Composites

Chitosan is a renewable, eco-friendly and biodegradable material that can be used as an alternative material for synthetic petroleum-based polymers. The use of biopolymers alone may be ineffective in removing the organic compounds from water. Its low mechanical strength, tensile strength, permeability and thermal stability had restrict its wide application [48]. Therefore, the best way to improve the properties such as increase stability of chitosan and its performance is to introduce reinforcement materials within the micro-regime or nano-regime [49].

Chitosan composites are biopolymer-based materials made up of two or more different substances that combine together to produce a new material with better performance than a single constituent material. These composite materials consist of two main phases: the matrix phase with low modulus and high elasticity, and the reinforcement phase with higher loading capacity [50]. A suitable biopolymer composite should be inexpensive, available in abundant, efficient, eco-friendly, biocompatible and reusable [51]. Typically, chitosan composites can be synthesised through crosslinking reaction between the positively charged group of chitosan and negatively charged crosslinking agent in the ionic gelation process.

### 3.1. Chitosan/Zeolite

Zeolite mainly contains aluminate and silicate with three-dimensional framework structures. It can be employed to remove organic pollutants due to its excellent adsorption properties with outstanding ion exchange and modification capabilities [52]. Natural zeolite is an abundant, environmentally friendly, inexpensive, chemically and mechanically stable hydrated alumina silicate material that can be used to form composites with chitosan. Xie et al. [53] have employed chitosan/zeolite composites to remove cationic, anionic and organic pollutants from aqueous solution with 31.6 mg/g adsorption capacity. Furthermore, Dehghani et al. [37] conducted an experimental study for the removal of methylene blue dye using pure chitosan and chitosan/zeolite composite. The results indicated that the dye removal efficiency using chitosan/zeolite was higher than pure chitosan. It was reported that 84.85% of the methylene blue was removed by the chitosan/zeolite composite with a maximum adsorption capacity of 24.5 mg/g.

The chitosan/zeolite composite was synthesised through the bridging mechanisms and the formation of hydrogen bonds between the surface functional groups of chitosan with the –OH, silano and aluminol groups of zeolite [52]. Based on the scanning electron microscopy (SEM) images obtained by Metin, Çiftçi and Alver [54], it was observed that pure chitosan had a lower specific surface area with a flake-like and smooth surface structure. However, the SEM image of chitosan/zeolite composite demonstrated a heterogeneous, irregular and rough surface as compared to pure chitosan. Besides, chitosan/zeolite possessed better thermal stability and experienced lower weight loss than pure chitosan during the thermal degradation process [55]. Therefore, the dye removal efficiency and thermal stability of chitosan were improved by immobilising zeolite in the chitosan matrix.

### 3.2. Chitosan/Carbon-Based Materials

Carbon-based materials such as multiwalled carbon nanotubes (MWCNTs) and graphene oxide are attractive adsorbents for water treatment. The adsorption capacity can be improved by introducing MWCNTs into the chitosan matrix. This is because MWCNTs have high surface areas, functionalisation capability with different surface functional groups and controllable size distribution. It was reported that chitosan/MWCNTs composite had higher thermal and mechanical strength as compared to pure biopolymer [56].

Salam, Makki and Abdelaal [57] added MWCNTs with average diameters of 60 to 100 nm in the chitosan matrix to synthesise a chitosan/MWCNTs composite. The MWCNTs were well dispersed on the surface of chitosan. Based on the SEM images, the membrane showed a porous structure morphology and sponge-like structure. Besides, TGA analysis showed that the thermal stability of chitosan/MWCNTs composite was higher than raw chitosan. The surface area of the MWCNTs was increased from 82.4 to 135.1 m^2^/g after the modification with chitosan. The excellent dispersion of MWCNTs in a chitosan matrix decreased the tangling and agglomeration of MWCNTs and greatly improved the adsorbent surface area.

Besides, graphene oxide has become a promising adsorbent material in wastewater treatment due to its excellent properties such as high surface areas and chemical stability. Graphene oxide consists of many functional groups, including –COOH, –OH, diol, epoxy, and ketone groups. These functional groups cause graphene oxide to have higher hydrophilic and able to compatible with other biopolymers [58]. The –COOH group of graphene oxide can react with the –NH_2_ group of chitosan to form chitosan/graphene oxide composites. Huyen et al. [59] prepared chitosan/graphene oxide composites by lyophilisation for the removal of methylene blue. The results showed that the chitosan/graphene composites were found to be a suitable adsorbent for methylene blue with a maximum adsorption capacity of 662.25 mg/g. After the adsorption, chitosan/graphene oxide composites could be easily separated and recovered by filtration.

### 3.3. Chitosan/Metal Oxides

There are many types of metal oxide nanoparticles such as zinc oxide (ZnO), magnesium oxide (MgO), magnetite (Fe_3_O_4_) and maghemite (γ-Fe_2_O_3_) that can be used to form chitosan-based composites. Among these metal oxides, ZnO is cheaper and has higher removal efficiency of organic pollutants than other metal oxides due to its higher semiconducting properties and high surface area [60]. Therefore, ZnO is commonly used to combine with chitosan to form chitosan/ZnO nanocomposite.

Chitosan/ZnO nanocomposite is a new hybrid material that can more efficiently remove organic pollutants than pure chitosan. Introducing ZnO nanoparticles into the chitosan matrix can increase the number of active sites on the chitosan surface, resulting in an improvement of the adsorption performance. On the other hand, the modification of chitosan with ZnO nanoparticles can provide a surface strengthening effect and improve surface corrosion resistance. An experimental study for the removal of organic pollutants was carried out by Arafat et al. [61]. The results showed that chitosan/ZnO nanocomposite could remove 95–99% of organic pollutants by using 2 mg/L of the composite at the temperature of 50 °C and contact time of 60 min.

Besides, MgO nanoparticles can be used for the removal of organic dyes due to their non-toxicity and high chemical stability [62]. However, the biosorption capacity of pure MgO was reported to be lower, but the contact time required to remove organic dyes was shorter than pure chitosan [63]. Therefore, Nga, Thuy Chau and Viet [64] introduced MgO nanoparticles in the chitosan matrix in order to remove reactive blue dyes with high biosorption efficiency in a short contact time. The results revealed that chitosan/MgO could remove up to 77.62% of the reactive blue dyes at the contact time of 120 min, which was shorter than the time required by pure chitosan. 

In addition, Fe_3_O_4_ and γ-Fe_2_O_3_ are considered iron oxide nanoparticles and they are also known as magnetic materials. They have been widely used to remove organic dyes from wastewater through magnetic separation processes [65]. This might be due to their high thermal stability, chemical stability and biocompatibility, as well as their good magnetic properties [66]. In order to improve the biosorption capacity of chitosan, Zhu et al. [67] and Azari et al. [68] incorporated Fe_3_O_4_ and γ-Fe_2_O_3_ in the chitosan matrix to form chitosan/γ-Fe_2_O_3_ and chitosan/Fe_3_O_4_/glutaraldehyde composites, respectively. The results indicated that magnetic chitosan composites exhibited higher biosorption capacity and could remove the organic dyes in a shorter contact time (<60 min) compared to pure chitosan (<180 min).

### 3.4. Chitosan/Other Biopolymers

Cellulose is a polysaccharide polymer that can be obtained abundantly from plants, and it is relatively cheap. It can be introduced into a chitosan matrix to form a biopolymer composite. Sun et al. [69] carried out an experimental study to synthesise a chitosan/cellulose composite using ionic liquids to remove organic pollutants from aqueous solution. Based on their results, the adsorption capacity of chitosan/cellulose composite was three times higher than the raw chitosan and cellulose. The chitosan/cellulose composite has a porous structure, larger surface areas, higher stability and higher affinity for organic pollutants [70].

Besides, Li et al. [71] had incorporated polyvinyl alcohol (PVA) in the chitosan matrix to synthesise chitosan/PVA composite as a biosorbent to remove organic dyes. Although PVA has exceptional physicochemical properties, it is not suitable to be utilised as an individual adsorbent because of its low solubility in water and low adsorption capacity for organic dyes [71]. However, chitosan can efficiently adsorb organic dyes due to the presence of a large amount of –OH and –NH_2_ groups on its surface. Hence, it is suggested to combine the properties of both PVA and chitosan to prepare chitosan/PVA composites. The composites will have high mechanical stability, high performance of adsorption and high chemical resistance in acidic and alkaline solution [41].

## 4. Biosorption Parameter Studies

The biosorption efficiency is affected by many factors, including solution pH, biosorbent dosage, initial pollutant concentration, contact time and solution temperature. However, the effects of these operating parameters are studied individually while maintaining the rest at the constant condition.

### 4.1. Effect of Solution PH

The biosorption capacity of chitosan composite as the biosorbent is highly dependent on the surface charge of chitosan composite and the type of organic dyes in the aqueous solution. The organic dyes can be classified into two types, which are cationic dyes with a positive charge and anionic dyes with a negative charge. It was reported that solution pH and point zero charge (pH_pzc_) of chitosan composites were the most critical parameters affecting the biosorption capacity [72]. They have a drastic impact on the surface charge of chitosan composite and ionisation of functional groups present on the chitosan composite surface.

pH_pzc_ can be defined as the pH value at which the net surface charge of chitosan composite is equivalent to zero [73]. Under acidic conditions at low pH, hydrogen ions (H^+^) are generated, while hydroxide ions (OH^−^) are found under alkaline conditions at high pH. If the solution pH is more than pH_pzc,_ the surface of chitosan composite will be negatively charged due to the deprotonation of surface functional groups by OH^−^ at high pH condition. This will favour the biosorption of cationic dyes through electrostatic attraction. Nevertheless, the positively charged surface of chitosan composite will occur at solution pH beyond pH_pzc_ due to the protonation of surface functional groups. This eventually results in strong electrostatic repulsion between cationic dyes and chitosan composite, thus reducing the biosorption efficiency. Vice versa in the case of anionic dyes, the favourable biosorption condition is attributed to the electrostatic attraction between the negatively charged dyes with the positively charged surface of chitosan composite.

For instance, Kausar et al. [74] reported that the pH_pzc_ for chitosan/clay composite was at pH 7.0. The biosorption capacity of chitosan/clay composite for Rose FRN dyes increased until pH 10. The results confirmed that the chitosan/clay composite was negatively charged since the optimal solution pH exceeded pH 7.0. Equation (1) shows that OH^−^ deprotonates the –COOH group on the composite surface and carboxylate ion (–COO–) with negatively charged and water (H_2_O) are formed under alkaline condition. Moreover, the biosorption mechanism for the removal of cationic dyes at high pH could be well explained by Equation (2). The negative surface charge of chitosan composites could facilitate electrostatic interaction with the cationic Rose FRN dyes. On the other hand, the –OH group on the chitosan composite could interact with the cationic dyes through hydrogen bonding. However, the biosorption capacity reduced after pH 10 might be attributed to the excessive OH^−^ reacted with cationic dyes through precipitation.
(1)$$-\mathrm{COOH}+{\mathrm{OH}}^{-}↔-{\mathrm{COO}}^{-}+H_{2}O$$
(2)$$-{\mathrm{COO}}^{-}+{\mathrm{dyes}}^{+}↔-\mathrm{COO}-\mathrm{dyes}$$

Besides, there are many types of cationic dyes, including methylene blue, malachite green, methyl violet, crystal violet and so on. A similar finding was reported for the biosorption of methylene blue using chitosan/zeolite, chitosan/bentonite, chitosan/activated carbon, chitosan/silica/ZnO and chitosan/clay composites [54,72,73,74,75]. The respective chitosan composite achieved a maximum biosorption capacity at high pH.

In addition, the biosorbents such as chitosan/Fe_3_O_4_/graphene oxide, chitosan/Fe_3_O_4_/glutaraldehyde and chitosan/ZnO composites were employed to remove crystal violet, malachite green and methyl violet, respectively [18,68,76]. It was observed that their removal efficiencies of cationic dyes also increased at high pH. However, the biosorption efficiency of cationic dyes was reduced at low pH. The low biosorption efficiency might be due to the competition of H^+^ with cationic dyes for the binding sites on the chitosan composite surface under acidic condition.

Next, Travlou et al. [77] studied the effect of pH for the biosorption of anionic reactive black dyes using chitosan/graphene oxide as the biosorbent. It was reported that the chitosan/graphene oxide exhibited a neutral charge surface when pH was at 6.8. The biosorption efficiency of reactive black increased significantly when solution pH was reduced from 12 to 2. Hence, chitosan/graphene oxide composite had a positively charged surface as the optimal solution pH below 6.8. At high pH, the high amount of OH^−^ in the aqueous solution increased the repulsive force between –COO– groups and anionic dyes, resulting in low biosorption efficiency of anionic dyes [78]. 

The similar findings also reported in the literature for chitosan/MWCNTs, chitosan/PVA, chitosan/cellulose and chitosan/quartzite composites [47,78,79,80]. The respective chitosan composite was used to remove reactive orange, reactive black, direct blue, eosin yellow and congo red, respectively. These anionic dyes consist of sulphonate (–SO_3_–) groups with negatively charged. When pH below pH_pzc_, –NH_2_ groups on the chitosan composite surface were protonated to form positively charged amine groups (–NH_3_^+^) under acidic condition. Consequently, the electrostatic interaction occurred between –NH_3_^+^ groups of chitosan composite with –SO_3_^−^ groups of anionic dyes as presented in Equation (3) [77].
(3)$$-{\mathrm{NH}}_{3}{}^{+}+-{\mathrm{SO}}_{3}{}^{-}↔-{\mathrm{NH}}_{3}{}^{\pm }O_{3}S$$

### 4.2. Effect of Biosorbent Dosage

The biosorbent dosage is an important factor in determining the optimum amount or saturation point. At saturation point, any further increment in the amount of biosorbent will not contribute to any significant improvement in the biosorption process. In general, introducing a high amount of biosorbent will increase the availability of vacant biosorption sites and also provide a large surface area for the biosorption of dye molecules [32]. Consequently, a high number of organic dyes can be adsorbed on the biosorbent active site, thereby increasing the removal efficiency of organic dyes.

For instance, the removal efficiency of reactive blue was improved by increasing the chitosan/MgO dosage from 2 to 14 g/L [64]. Nevertheless, it was reported that the removal efficiency increased slightly from 58.70% to 59.82% when the biosorbent dosage was further increased to 16 g/L. Hence, it concluded that 14 g/L was considered as the optimum biosorbent dosage. The results indicated if the biosorbent dosage exceeded the optimum biosorbent dosage, it would not provide any improvement in the biosorption process due to the unsaturated biosorbent active sites. 

Similarly, the biosorption efficiency of methylene blue also increased from 52.52% to 76.78% when the chitosan/carbon clay composite was increased from 0.1 to 0.2 g/L [81]. However, the biosorption efficiency eventually decreased to 77.24% when the dosage was 0.3 g/L. Therefore, the optimal dosage of chitosan/carbon clay composite was determined to be 0.2 g/L. This might be caused by the overlapping or agglomeration of excessive biosorbent particles that would block the binding sites of another biosorbent. Introducing excessive biosorbent in the biosorption process resulted in an equilibrium imbalance between the vast number of biosorbent binding sites and a constant number of dye molecules [57]. As a result, the biosorption capacity was reduced due to the decrease in the surface area of chitosan composite and the availability of biosorption sites. A similar finding was also reported for chitosan/PVA and chitosan/activated carbon composites [47,82]. The biosorption efficiency of anionic dyes reduced when the biosorbent dosage was added in excess amounts.

### 4.3. Effect of Initial Dye Concentration

The initial concentration of dyes is also a major influencing factor in the biosorption process which can be studied through biosorption isotherm models. The effect of initial dye concentration on biosorption capacity of chitosan/kaolin clay composite was studied at a constant time of 400 min with the initial dye concentration studied in the range of 50–400 mg/L [83]. The results revealed that the biosorption capacity increased with an increase in the initial dye concentration. At the initial biosorption process, more binding sites on the chitosan composite surface were available for the biosorption of dye molecules. Besides, the high initial concentration of dyes provided a greater driving force required to overcome the mass transfer resistance between the liquid phase and solid phase [84]. It could increase the mass transfer of organic dyes to binding sites across the boundary layer of biosorbent particles against the concentration gradient.

However, the biosorption capacity of chitosan composite decreased and remained constant after a period of time. This might be due to the saturation of chitosan composite surface, where dye molecules have occupied most binding sites [71,83]. As a result, excessive dye molecules were not adsorbed on the binding sites due to saturation of binding sites, thus reducing the biosorption capacity. Similar findings also reported in the literature for chitosan/PVA, chitosan/activated carbon and chitosan/MgO composites [64,71,85]. The respective composite also exhibited high biosorption efficiency as increasing the initial dye concentration.

### 4.4. Effect of Contact Time

Contact time is one of the most significant parameters as it can be employed to estimate the biosorption equilibrium and biosorption kinetic models. The contact time can be defined as the time given for the immersion of a given amount of biosorbent at a constant volume and concentration of organic dyes in the solution [86]. According to Muinde et al. [76], the effect of contact time on the biosorption efficiency of cationic dyes was studied at a constant initial dye concentration of 2.3 mg/L by using chitosan/ZnO composite as the biosorbent. Based on the results, the biosorption rate was significantly accelerated during the first 15 min with a deeper gradient. This might be due to the abundance of active sites available on the biosorbent surface that could be easily accessed by the organic dyes. Over a period of time, a large number of organic dyes was adsorbed on the active sites of the chitosan composite surface through intermolecular interaction. Muinde et al. [76] also found that the biosorption rate decreased until an equilibrium state was achieved at 180 min with the maximum biosorption efficiency of 80.10%. The decrease in concentration gradient with contact time could be attributed to a decrease in vacant binding sites for organic dye biosorption. The remaining organic dyes in the solution competed with each other to occupy the remaining active sites on the biosorbent surface. Therefore, sufficient time should be ensured to establish the solid-liquid equilibrium [84]. Besides, Rangabhashiyam, Anu and Selvaraju [85] and Nga, Thuy Chau and Viet [64] also discovered similar findings in which the biosorption rate decreased gradually as contact time increased.

### 4.5. Effect of Solution Temperature

Solution temperature is considered an important parameter that will affect the diffusion rate of sorbates via the external boundary layer surrounding the biosorbent. The information obtained from effect of solution temperature can determine whether the nature of the biosorption process is endothermic or exothermic [87]. In general, the biosorption process at high solution temperatures can increase the collision between the biosorbent and sorbate more frequency, thereby enhancing the biosorption rate. In addition, an increase in solution temperature will decrease the thickness of the boundary layer surrounding the biosorbent. Consequently, the mass transfer resistance of sorbates across the boundary layer is reduced. This will eventually increase the diffusion rate of dye molecules moving from the aqueous phase to the biosorbent surface.

In the research work conducted by Travlou et al. [77], the chitosan/graphene oxide composite was used as the biosorbent to remove organic dyes. The effect of solution temperature was investigated in the range of 30 to 60 °C. It was reported that the removal efficiency of organic dyes increased when the solution temperature was raised from 30 to 45 °C. The result indicated the dye biosorption was an endothermic process that required a high amount of heat supplied to the process. However, the removal efficiency reduced as the solution temperature was further increased to 60 °C. The excessive heat supplied to the biosorption might lead to the deactivation or the destruction of binding sites on the biosorbent surface [44]. Therefore, the removal efficiency decreased at high solution temperature.

Nevertheless, Kausar et al. [74] discovered a different finding where the biosorption capacity of chitosan/clay decreased with increasing solution temperature from 30 to 60 °C. The result suggested that the biosorption of direct Rose FRN dye was an exothermic process that released heat to the surrounding. Hence, an increase in solution temperature would supply a high amount of heat to the biosorption process, which could deactivate the binding site on the chitosan composite surface. Furthermore, Bahrudin, Nawi and Sabar [88] also observed a similar decreasing trend in the biosorption capacity of chitosan/ montmorillonite when increasing the solution temperature.

### 4.6. Comparison Performance for Various Types of Chitosan Composites

Table 2 compares the performance for various types of chitosan composites used to remove cationic dyes and anionic dyes under different optimum operating conditions. The results revealed that the optimal pH range for the removal of cationic dyes was between 8 and 11. As for the removal of anionic dyes, the optimal pH range was found to be between 2 and 6.8 under acidic condition. In addition, all of the biosorption processes were operated in the temperature range of 25–45 °C. It is difficult to compare the performance of chitosan composite accurately due to the differencce in all the operating conditions. However, the biosorption efficiency or organic dyes was mainly affected by solution pH, biosorbent dosage, initial pollutant concentration, contact time and solution temperature as discussed earlier.

Among all the parameters studied, solution pH contributed the significant effects on the dye removal efficiency. This is due to the solution pH could affect the surface charge of chitosan composite and the type of dyes, eventually affecting dye removal efficiency. The results suggested that the dye removal process was chemisorption [37,79,90], which involved electrostatic interaction and ions exchange between chitosan composite and dye molecules. Besides, the physisorption process depended on the binding sites provided by chitosan composite and the amount of dyes adsorbed on the binding sites [89,91]. The physisorption process was mainly affected by biosorbent dosage, initial dye concentration, contact time and solution temperature. 

## 5. Biosorption Mechanism

The biosorption process is a mass transfer process that moves substances from the liquid phase (water) to the solid phase (biosorbent). Biosorption can be referred to as the adsorption of sorbates such as atoms, molecules or molecular ions on the surface of solid biological sorbent [92]. Biosorption continues until an equilibrium is reached where the amount of organic dyes adsorbed on the surface of the biosorbent is the same as the amount of organic pollutants left in the solution [51]. In general, it is a physicochemical and metabolism independent process to remove substances from solution by biological materials. 

In recent years, the application of biological materials to adsorb and remove organic pollutants from water bodies has attracted a great deal of attention. It has become a hot topic among researchers due to the problems and shortcomings faced by the conventional methods in removing non-biodegradable organic pollutants. Biosorption is characterised by remarkable merits such as ease of modification of biosorbents, high efficiency and low operating and maintenance costs. Another significant benefit of this process is that it does not lead to the generation of intermediate products [93]. The process is reversible, and the adsorbent used can be regenerated by desorption for reuse.

This process can effectively remove organic pollutants even when organic pollutants are in low concentration. This is achieved through the binding of organic pollutants onto the vastly available active binding sites present on the surface of the biosorbent. The detailed biosorption mechanisms were discussed in this section. 

The high efficiency of biosorption is a vital aspect that should be characterised to the biosorbent for the effective elimination of various organic dyes. The physicochemical features of the biosorbent vary due to the presence of the different functional groups with varying degrees present on its surface. The most common functional groups present on the biosorbent surface are –NH_2_, –COOH, –OH and phosphate group. This respective functional groups can facilitate the sorption of organic dyes on the biosorbent through multiple sorption mechanisms. Figure 4 illustrates the different types of mechanisms involved in the biosorption process. In fact, the interaction between organic dyes and biosorbent surface occurs through electrostatic interaction, ion exchange, complexation, chelation and microprecipitation, physical and chemical adsorption [84].

The biosorption interaction between sorbate and biosorbent mainly occurs in two different conditions: surface sorption and interstitial sorption. Surface sorption involves film diffusion. It occurs when sorbate moves through the bulk solution and diffuse across the liquid film boundary layer surrounding the biosorbent surface. The biosorbent provides numerous active sites for sorbate binding. After that, sorbate is adsorbed on the opposite charged of binding sites on the biosorbent surface [94]. This phenomenon is strongly promoted by Van Der Waals forces, dipole interactions or hydrogen bonding [95]. This process is followed by interstitial sorption. During interstitial sorption, it involves intraparticle diffusion. Sorbate further diffuses into the pores of the biosorbent and eventually attaches to the interior surface of the biosorbent. In this process, film diffusion and intraparticle diffusion are considered as the rate-determining steps whereas the surface bonding takes place in a faster rate [51]. The rate-determining step is defined as the kinetic process occurs at the slowest rate, which eventually affects the overall biosorption rate [79]. As a result, organic dyes require a longer contact time to diffuse from the aqueous phase and internal surface of biosorbents by passing through the biosorbent pores [27,84]. Therefore, the rate-determining steps must be determined in order to improve the overall biosorption rate.

## 6. Biosorption Kinetics

The biosorption kinetic studies are very useful in determining the biosorption rate of dye molecules on the surface of the chitosan composite and the time required to achieve biosorption equilibrium [96]. The kinetic studies are conducted to identify the rate-limiting step and biosorption mechanisms such as mass transfer, diffusion and chemical reaction. Several mathematical models have been employed in the literature to investigate kinetic studies.

In a recent analysis, the fitness of experimental data of dyes removal by chitosan composite was examined using two common kinetic models namely, pseudo-first-order (PFO) and pseudo-second-order (PSO) kinetic models. These kinetic models can evaluate whether the biosorption process involves physisorption or chemisorption as well as investigate the reaction order. The corresponding kinetic models are expressed in Equation (4) and Equation (5) [8,79]. The PFO kinetic model reveals that the rate of dye biosorption is proportional to the amount of dye adsorbed on the chitosan composite. The PSO kinetic model shows that the biosorption rate is proportional to the square of the amount of dyes adsorbed on the biosorbent [97].

PFO model linear equation,
(4)$$\log\left(q_{e}-q_{t}\right)=\log\left(q_{e}\right)-\left(\frac{k_{1}}{2.303}\right)t$$

PSO model linear equation,
(5)$$\frac{t}{q_{t}}=\frac{1}{k_{2}q_{e}{}^{2}}+\left(\frac{1}{q_{e}}\right)t$$
where
*q_e_* = mass of dyes adsorbed per unit mass of chitosan composite at equilibrium condition, mg/g*q_t_* = mass of dyes adsorbed per unit mass of chitosan composite at any time *t*, mg/g*t* = contact time, min*k_1_* = rate constant of PFO model, min^−1^*k_2_* = rate constant of PSO model, g/(mg·min)

The selection of the kinetic models mainly depends on the fitness of the experimental data based on the coefficient of determination (R^2^) value. Besides, the calculated biosorption at equilibrium value should be close to the experimental biosorption at equilibrium value [98]. If the experiment data is well fitted to the PFO model with a R^2^ value close to 1, it suggests that physical sorption will be the rate-limiting step of the biosorption. Vice versa in the case of the PSO model, the rate-limiting step is the chemisorption. In general, the biosorption rate constants and biosorption at equilibrium value of both kinetic models are computed based on the slope and intercept of the graph, respectively. The slope and the intercept of the linear plot of log (*q_e_* – *q_t_*) against *t* are used to predict the rate constant and biosorption at equilibrium values for the PFO model. As for the PSO model, the rate constant and biosorption at equilibrium values are determined based on the slope and intercept of the linear graph of *t*/*q_t_* against *t* [98,99].

The kinetic data for the removal of various organic dyes using different types of chitosan composites obtained from the literature are summarised in Table 3. It could be observed that all the biosorption processes studied in the literature were well fitted to the PSO kinetic model with a higher R^2^ value than the PFO model. Furthermore, the value of biosorption at equilibrium calculated from the PSO model demonstrated a smaller difference with the experimental biosorption at equilibrium value than those calculated from the PFO model. Therefore, these findings suggested that the biosorption of dyes was better defined by the PSO model instead of the PFO model. The findings indicated that the biosorption of dye molecules was dominated by chemical sorption, which involved the sharing of valence forces or electron exchange between the dyes and the active sites on the surface of chitosan composite [79].

Among the chitosan composites listed in Table 3, chitosan/cellulose composite demonstrated the highest experimental biosorption at equilibrium value of 381.70 mg/g. This might be attributed to the high availability of surface functional groups on chitosan/cellulose such as –OH and –COOH groups provided by the cellulose that facilitated the chemical interaction with dye molecules. According to the SEM analysis conducted by Wang et al. [80], the result indicated that chitosan/cellulose composite showed an internal structure with highly porous and multiple membrane layers. Therefore, the high porosity of chitosan/cellulose composite contributed to a large adsorption site for the biosorption of dyes, resulting in high biosorption capacity.

Since PFO and PSO models could only be employed to evaluate the type of biosorption, therefore the diffusion mechanism of the dye biosorption was studied by applying the intraparticle diffusion model. The respective model was developed by Weber and Morris and expressed in Equation (6) [37,82]. The intraparticle diffusion model can determine the type of biosorption mechanisms that limits the biosorption rate.

Intraparticle diffusion model linear equation,
(6)$$q_{t}=k_{p}t^{0.5}+C$$
where
*k_p_* = rate constant of intraparticle diffusion, mg/(g·min^0.5^)*C* = y-intercept of the graph that related to the boundary layer thickness, mg/g


Generally, the biosorption process was assumed to take place in these three consecutive steps: film diffusion, intraparticle diffusion and biosorption [94]. Firstly, it involved the external mass transfer of dye molecules from the bulk solution across the boundary layer to the external surface of the chitosan composite. Secondly, it involved intraparticle diffusion, where the dye molecules diffused from the external surface to the internal surface of chitosan composite by passing through the pores. Lastly, it was followed by the biosorption of dye molecules on the binding sites of chitosan composite until achieving equilibrium [102].

By plotting the linear graph of *q_t_* against *t^0.5^*, the values of the rate constant of intraparticle diffusion and y-intercept can be obtained from the slope and intercept of the graph. For example, Figure 5 demonstrates the linear plot of *q_t_* against *t^0.5^* of intraparticle diffusion model for the biosorption of Reactive Blue 5 dyes using chitosan/quartzite composite [80]. Figure 5 shows the biosorption process comprised three sections of the linear line. The first section with a deeper gradient was attributed to the effect of the boundary layer. The organic dye molecules were transported through film diffusion at a high diffusion rate to the external surface of chitosan/cellulose composite. Once the external surface was saturated, the organic dye molecules further diffused through the pore and reached the internal surface of chitosan/cellulose. This phenomenon was resulted from the intraparticle diffusion effect and was represented by the second section with a smaller gradient. Lastly, the equilibrium adsorption of dye molecules on the binding site of chitosan composite occurred in the third section of the linear line.

Similar findings also reported in the literature on the synthesis of chitosan composite with the incorporation of ZnO, γ-Fe_2_O_3_/silica-oxide, zeolite, PVA and activated carbon [37,67,82,100,103]. After conducting the literature studies, the values of the rate constant of intraparticle diffusion and y-intercept for various chitosan composites are tabulated in Table 4. According to the model, if the intraparticle diffusion is the rate-limiting step in the dye biosorption process, the linear line should intercept the origin. However, none of the model lines for the investigated chitosan composites passed through the origin. The greater y-intercept value implied that the biosorption process was influenced not only by the intraparticle diffusion but also by a certain degree of boundary layer [97]. In short, the greater the value of y-intercept, the more significant the boundary layer effect.

## 7. Biosorption Isotherms

The biosorption isotherms are widely used to investigate the interaction type between dye molecules and chitosan composites, as well as to estimate the maximum biosorption capacity. There are two common biosorption isotherm models such as Langmuir and Freundlich isotherm models, which have been employed in the literature to determine the mass of dye molecules adsorbed onto the chitosan composite surface and the equilibrium concentration of dyes [59,94].

The Langmuir isotherm model assumes that the biosorption occurred on the homogenous surface of chitosan composite where the organic dye molecules adsorbed on a constant number of chitosan composite binding sites [91]. Once the binding sites of chitosan composite were saturated, no further biosorption will occur as the maximum biosorption capacity was achieved. A monolayer of molecular thickness was formed on the chitosan composite surface with no interaction between the organic dye molecules. Besides, the adsorption energy and enthalpy were assumed to be distributed equivalent to chitosan composite binding sites [78]. The Langmuir isotherm model can be expressed in non-linear and linear as presented in Equation (7) and Equation (8), respectively [8,79].

Langmuir isotherm model in non-linear form,
(7)$$q_{e}=\frac{q_{max}K_{L}C_{e}}{1+K_{L}C_{e}}$$

Langmuir isotherm model in linear form,
(8)$$\frac{C_{e}}{q_{e}}=\frac{1}{q_{max}K_{L}}+\left(\frac{1}{q_{max}}\right)C_{e}$$
where
*C_e_* = equilibrium concentration of dyes, mg/L*q_max_* = maximum biosorption capacity, mg/g*K_L_* = Langmuir constant, L/mg

Besides, another important Langmuir constant in the Langmuir isotherm model—namely, separation factor—can be employed to determine whether the dye biosorption is favourable, non-favourable, linear or irreversible [104]. If the value of separation factor falls within the range between zero and one, it is considered a favourable biosorption process. Otherwise, the biosorption process is considered non-favourable when the separation factor value obtained is more than one. Meanwhile, the biosorption is considered linear when the separation factor value is equal to one and irreversible when the separation factor value is equal to zero. The mathematical representation of separation factor is presented in Equation (9).

Separation factor of Langmuir isotherm model,
(9)$$R_{L}=\frac{1}{1+K_{L}C_{O}}$$*R_L_* = separation factor *C_o_* = concentration of dyes at initial process, mg/L*K_L_* = Langmuir constant, L/mg

In contrast, the Freundlich isotherm model assumed that the biosorption took place on the heterogeneous surface of chitosan composite with the interaction between multilayer adsorption of dye molecules [90]. It also assumed the adsorption energy was non-uniformly distributed on the chitosan composite surface [78]. The non-linear and linear form of Freundlich isotherm model equations are expressed in Equation (10) and Equation (11), respectively [8,79]. The heterogeneity factor is significant in analysing the type of biosorption involved in the biosorption process. The biosorption process is linear when the value of heterogeneity factor is equal to one. Conversely, when heterogeneity factor is less than one or greater than one, the removal of organic dyes involves either chemisorption or physisorption.

Freundlich isotherm model in non-linear form,
(10)qe=KFCe12

Freundlich isotherm model in linear form,
(11)$$logq_{e}=logK_{F}+\frac{1}{n}logC_{e}$$
where*n* = heterogeneity factor*K_F_* = Freundlich constant related to the adsorption capacity, (mg/g)(L/g)^1/n^

The linear plots for both isotherm models are established using the linear equation instead of the non-linear equation so that the adsorption isotherm parameters can be determined easily from the slope and intercept of the graph. For instance, the isotherm parameter values such as Langmuir constant and maximum biosorption capacity for the Langmuir isotherm model are obtained by plotting *C_e_/q_e_* against *C_e_*. The slope and intercept of the respective graph represent Langmuir constant and maximum biosorption capacity, respectively. For the Langmuir isotherm model, the heterogeneity factor and Freundlich constant values can be computed from the slope and intercept of the linear plot, respectively, by plotting log *q_e_* against *log C_e_*.

Table 5 tabulates and summarises the adsorption isotherm parameters of Langmuir and Freundlich isotherm models for the biosorption of dyes using different types of chitosan composites as biosorbents. The results revealed that most of the biosorption process fitted into the Langmuir isotherm model with a higher R^2^ value than the Freundlich isotherm model. Langmuir isotherm model suggested that a monolayer of dye molecules was formed on the homogeneous surface of chitosan composite and no interaction between the dye molecules. The interaction between dye molecules and chitosan composite was stronger, reflecting that it was chemisorption. Since all of the Langmuir constant values were in the range between zero and one, this indicated that it was a favourable biosorption process.

However, some researchers found that the biosorption processes using chitosan/quartzite [78], chitosan/zeolite [37] and chitosan/MgO [64] as biosorbents followed the Freundlich isotherm model with a higher R^2^ value than the Langmuir isotherm model. The findings implied that the biosorption of dyes occurred on the heterogeneous surface. The interaction occurred between the dye molecules with chitosan composite and between the adsorbed dye molecules, resulting in the formation of multilayer molecular thickness. This was due to the presence of impurities adsorbed on the chitosan composites during the modification of pure chitosan [59]. The impurities on the chitosan composite surface would cause different level of binding energy for the biosorption of organic dyes. The calculated heterogeneity factor values were more than one which indicated the removal of dyes involved physisorption.

For the comparison of maximum biosorption capacity values, chitosan/kaolin clay composite possessed the highest maximum biosorption capacity value of 560.90 mg/g among the chitosan composites listed in Table 5. This might be due to the availability of –OH, silanol, and aluminol function groups on the surface chitosan/kaolin clay composite, which was attributed by kaolin clay [106]. These additional surface functional groups corresponded to cellulose served as adsorption sites to interact electrostatically with the organic dye molecules and maximise the biosorption capacity. The rough and irregular surface of chitosan/kaolin clay composite also helped to promote the biosorption of dyes.

In contrast, chitosan/zeolite composite exhibited the lowest maximum biosorption capacity value of 24.51 mg/g for the removal of methylene blue dyes as compared to the other chitosan composites. As suggested by the Freundlich isotherm model, the biosorption of methylene blue occurred on the heterogeneous surface of chitosan/zeolite composite. The removal of methylene blue was dominated by the physisorption process rather than the chemisorption process. Hence, the weak intermolecular interaction between the biosorbent and organic dye molecules, as well as between the dye molecules, might be broken easily. This might be attributed to the collision between dye molecules as the dye concentration and solution temperature increased [53]. Consequently, the dye molecules were desorbed from the chitosan/zeolite surface and remained in the aqueous solution, resulting in lower adsorption capacity.

## 8. Biosorption Thermodynamics

Thermodynamic studies are usually carried out to investigate the effect of temperature on the dye biosorption process. Besides, the studies also provide detailed insight information on the feasibility, spontaneity and nature of the biosorption process as well as the orderliness of organic dyes molecules at the solid-liquid interface [107]. This information can be obtained by determining the thermodynamic parameters such as Gibb’s free energy change ($\Delta G^{{}^{\circ}}$ in kJ/mol), enthalpy change ($\Delta H^{{}^{\circ}}$ in kJ/mol) and entropy change ($\Delta S^{{}^{\circ}}$ in J/(mol·K)). The corresponding thermodynamic parameters can be computed by using the mathematical formulas from Equation (12) to Equation (14) [103,108,109].
(12)$$K_{C}=\frac{C_{s}}{C_{e,m}}$$
(13)$$\Delta G{}^{\circ}=-RT\;\ln\;K_{c}$$
(14)$$\ln\;K_{F}=\frac{\Delta S{}^{\circ}}{R}-\frac{\Delta H{}^{\circ}}{RT}$$
here
*C_s_* = equilibrium concentration of dyes remaining in aqueous solution, mg/L*C_e,m_* = mass of dyes molecules adsorbed per unit mass of chitosan composite at equilibrium, mg/g*R* = gas constant (8.314 J/(mol·K))*T* = absolute temperature, K 

For instance, Gibb’s free energy change value is calculated by substituting Equation (12) into Equation (13) once the equilibrium constant (*K_c_*) has been determined. Next, Equation (14) shows the van’t Hott equation that can be used to calculate the energy change of the biosorption process. The value of enthalpy change and entropy change can be obtained from the slope and intercept of the graph by plotting ln *K_c_* against *1/T.*

The thermodynamic parameters for the biosorption of dyes using various chitosan composite were collected from different literature and summarised in Table 6. The enthalpy change is generally utilised to evaluate whether the nature of biosorption is an exothermic or endothermic process. The biosorption of dyes is an exothermic process if the measured enthalpy change is negative value; otherwise, it is an endothermic process [110].

Next, the biosorption process is considered favourable and spontaneous when the obtained Gibb’s free energy change is in negative magnitude. Besides, Gibb’s free energy change is also helpful in determining whether the biosorption is occurred either physically or chemically. If the value of Gibb’s free energy change lies between −20 kJ/mol and 0 kJ/mol, this indicates the type of biosorption is physisorption. However, if the Gibb’s free energy change value is within the range of −80 kJ/mol to −400 kJ/mol, the biosorption of dyes involves chemisorption [74]. Lastly, the orderliness at the solid-liquid interface increases with the increasing entropy change value.

Based on the results as shown in Table 6, it was observed that all the biosorption processes using various chitosan composites had the negative value of Gibb’s free energy change, reflecting that the biosorption processes were feasible and spontaneous at the low temperature range between 288 and 320 K. In addition, most of the organic dye biosorption processes was physisorption type since their Gibb’s free energy change values were within the range of −20 to 0 kJ/mol. The Gibb’s free energy change values of chitosan/quartzite, chitosan/activated carbon and chitosan/MgO composites were −25.88, −23.42 and −21.73 kJ/mol, respectively, which were in the range of −20 and −80 kJ/mol. These values indicated that the physical adsorption was influenced by certain chemical effects and implied that ion exchange might occur during the biosorption process [111,112].

Moreover, most of the biosorption processes with a positive value of enthalpy change revealed that it was an endothermic process that required input energy supplied to the process. The endothermic nature of organic dye biosorption might be attributed to the dehydration process, where additional energy was required to remove the water molecules adsorbed on the surface of chitosan composite. Meanwhile, the desorption of water molecules could provide a larger surface area for the biosorption of organic dyes on the chitosan composite surface [37]. Among the chitosan composite, chitosan/cellulose had the highest positive enthalpy change value due to the membrane layer of cellulose entrapped with a high amount of water molecules, thereby increasing the energy for the removal of water molecules. 

However, chitosan/PVA, chitosan/alumina and chitosan/γ-Fe_2_O_3_/silica oxide composites as the biosorbents for the organic dye removal demonstrated a different finding with a negative enthalpy change value. The finding suggested that the biosorption process was exothermic, where the organic dye molecules interacted with the surface functional groups of chitosan composites through electrostatic attraction, hydrogen bonding and weak intermolecular interaction. Consequently, the surface energy of chitosan composite was reduced and the output energy was released to the surrounding [8].

Furthermore, the biosorption process with a positive magnitude of entropy change reflected that the dye molecules had a good interaction with the chitosan composite. Consequently, this resulted in an increase in the irregularity of dye molecules at the solid-liquid interface. Conversely, the irregularity of dye molecules decreased when a negative entropy change was obtained [67,101].

## 9. Future Prospects and Conclusions

This work reviewed the results achieved by numerous researchers on the removal of organic dyes through biosorption process. The aim of this review is to present the potential of utilisation chitosan composites as low-cost adsorbent for textile wastewater treatment and to attract more research on large scale applicability. Biosorption using chitosan-based adsorbent stands out as one of the most attractive organic dyes removal methods in terms of cost-benefit and efficient performance. The utilisation concept of converting waste to wealth can promote sustainability in the wastewater treatment research field.

Chitosan-based adsorbent with higher specific surface area, appropriate pore size and pore volume and multiple functional groups such as hydroxyl, carboxyl and amine groups are beneficial for the effective and efficient removal of organic dyes. This article presented the biosorption parameter study, mechanisms involved in the processes, addressing kinetics, isotherms and thermodynamics. Factors such as solution pH, biosorbent dosage, initial dye concentration, contact time and solution temperature are those significant parameters that affect the biosorption efficiency. Modelling revealed that most of biosorption studies described by the PSO model, while the Langmuir and Freundlich isotherm equations have been successfully applied to many adsorption processes. Thermodynamics data demonstrated that biosorption of organic dyes using various types of chitosan composites are spontaneous and endothermic processes. The high molecular weight chitosan with thermally and chemically stable properties often encountered aggregation and polymeric chain entanglements during adsorption. Hence, recent research also focusses on the preparation of low molecular weight of chitosan, to improve the solubility of chitosan and to expand its potential applications in more extensive fields.

Even though the biosorption process is an efficient and cost-effective technique for the removal of organic dyes, it experiences a little more challenging to demonstrate it at large-scale commercial applications which would involve a significant financial and technological effort. Most of the chitosan biosorbents are in used the suspended form and the post-separation of suspended biosorbent from the treated effluent was extremely difficult, which are not effective and durable for a long-term environmental application. This would cause problems related to the recyclability, regeneration and maintenance of the used biosorbent stability, which restrict its application in the suspended form for the removal of organic dyes from wastewater. Limited number of investigations had been conducted in the large-scale application for biosorption of organic dyes in wastewater. Almost all the research works were performed in a laboratory batch scale, indicating a long journey ahead before implementation of the biosorption technology in real industry wastewater.

In order to overcome the drawbacks associated with the suspended form of the biosorbents, a greater number of studies need to be focused on the developing of immobilised, magnetic chitosan biosorbents or fixed-bed dynamic tests. These types of biosorbents can be separated out easily and/or recovered from treated medium after the biosorption process within a short time. The recovery and regeneration through desorption is a critical to evaluate the biosorbent recyclability together with the adsorption-desorption cycles, which aimed to be used in commercial/industrial scale later. A well-planned regeneration and management of biosorbent can reduce environmental impact and clogging issue, as well as enhance/maintain its stability, mechanical strength and bioremediation capacity.

Biosorption of real textile wastewater in fixed-bed column or continuous systems need to be conducted to evaluate the actual removal efficiency of a biosorbent and account for the multi-solute scenarios present in the real industry wastewater. The main criteria such as high adsorption rate and capacity, effectiveness separation, robustness and stability also need to be considered when designing the biosorbent, so that the biosorbent can be applied in a hash condition with minimal/ no leaching of harmful materials to the environment. Biosorbent surface need to be modified properly to strengthen the surface functional groups and improve the process efficiency of organic dyes. The supply chain for the raw materials of biomass-based chitosan biosorbent needs to be increased and explored, while nurturing the local expertise for effective chitosan production. It is also important to study the life cycle assessment on biosorption process and adsorbent in order to evaluate the impacts towards environment and human beings. An economic assessment of the chitosan-based in a large-scale biosorption system that aims at industrial application is an essential activity to ascertain and assess the feasibility of its future in wastewater treatment.

## Figures and Tables

**Figure 1 polymers-13-03009-f001:**
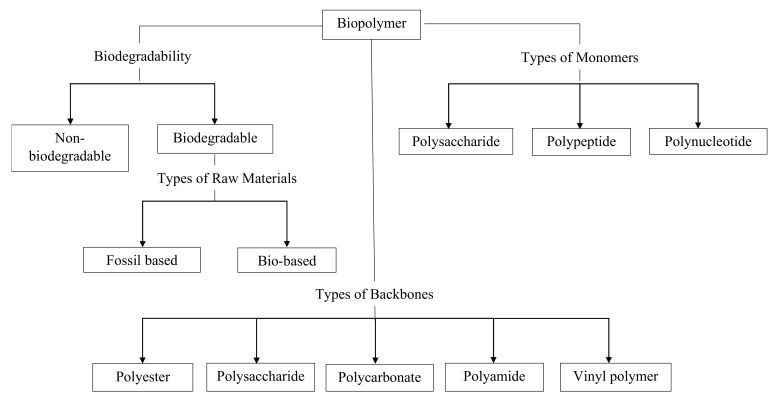
Classification of biopolymers based on the type of monomers, biodegradability, raw materials, and backbones [26]. Reproduced with permission from Elsevier Science Ltd.

**Figure 2 polymers-13-03009-f002:**
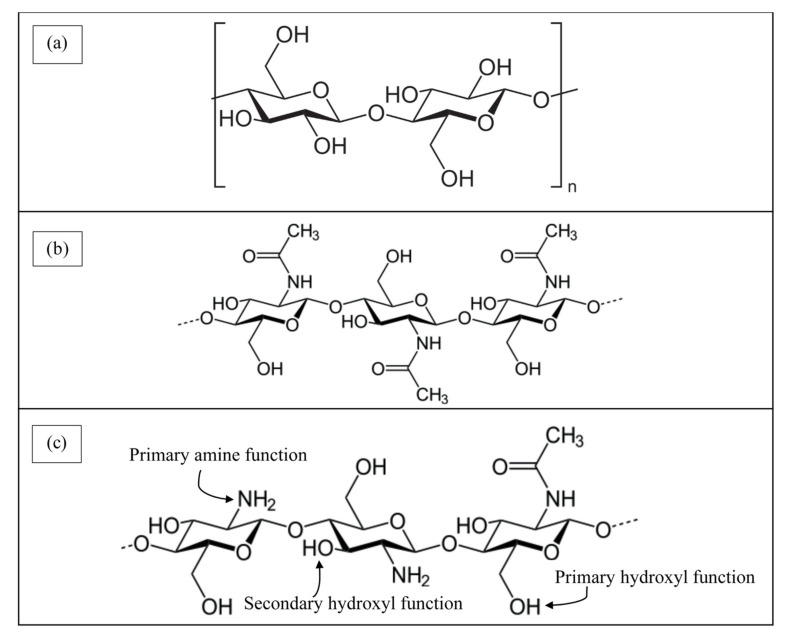
Chemical structures of (**a**) cellulose, (**b**) chitin and (**c**) chitosan.

**Figure 3 polymers-13-03009-f003:**
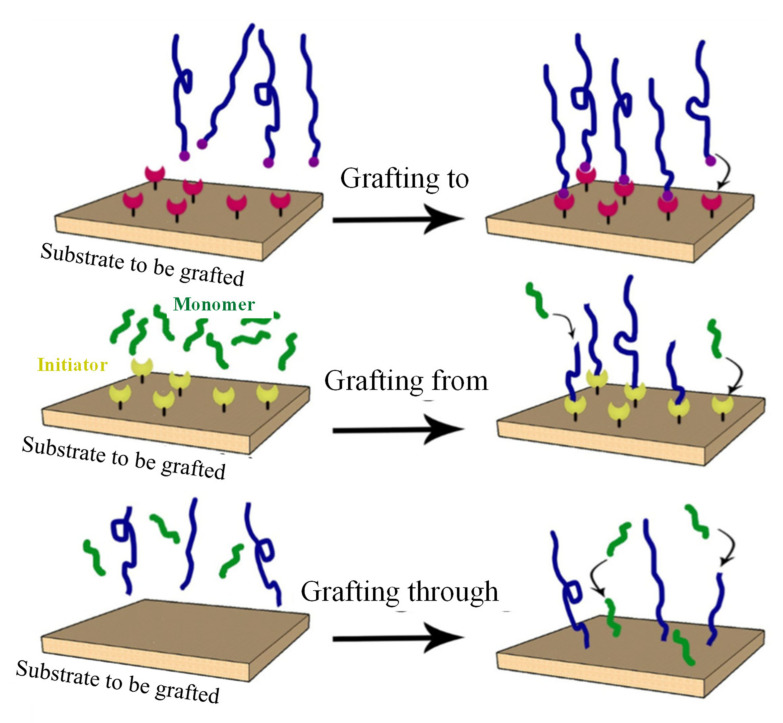
Polymer grafting methods [41]. Reproduced with permission from Elsevier Science Ltd.

**Figure 4 polymers-13-03009-f004:**
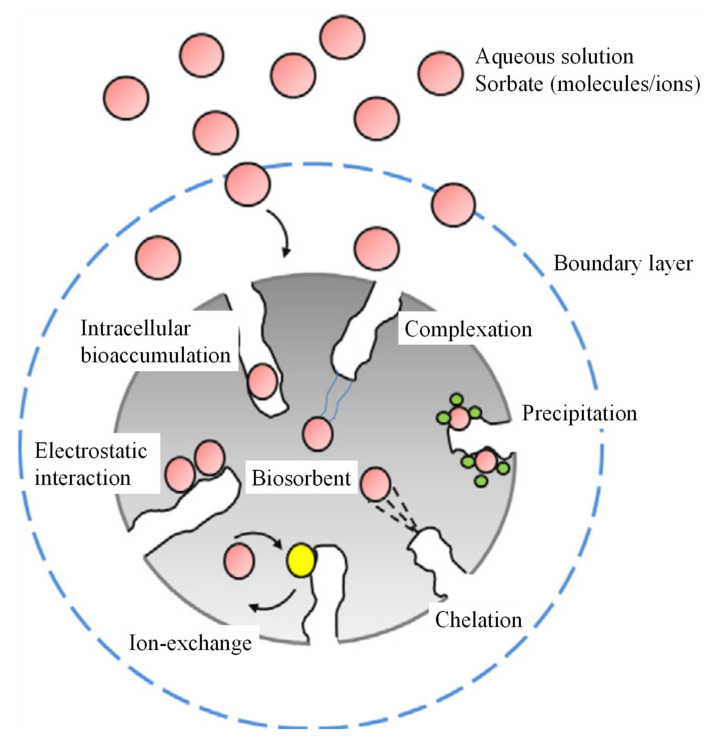
Types of mechanisms in biosorption [84]. Reproduced with permission from Elsevier Science Ltd.

**Figure 5 polymers-13-03009-f005:**
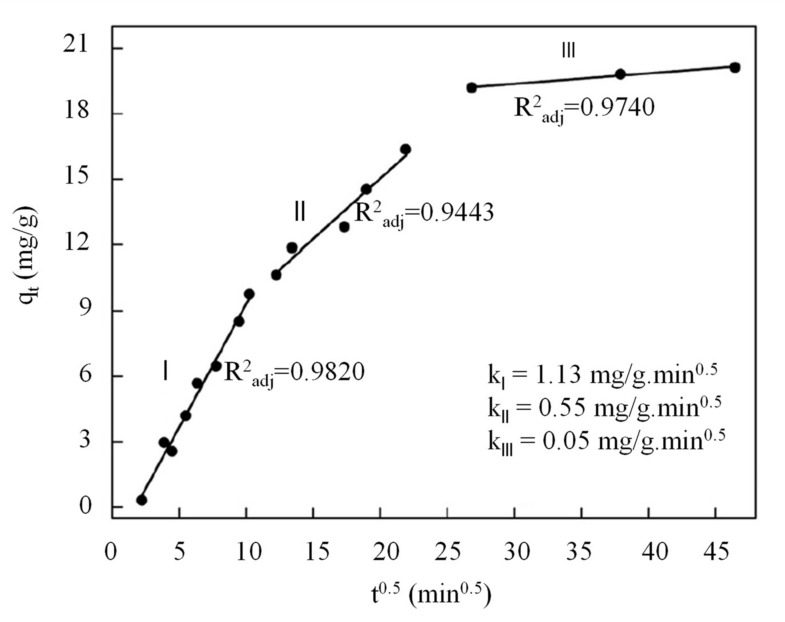
Linear graph of intraparticle diffusion model for biosorption of Reactive Blue 5 by chitosan/quartzite composite [78]. Reproduced with permission from Elsevier Science Ltd.

**Table 1 polymers-13-03009-t001:** Degree of fixation and percentage loss in effluent for various types of dyes [6].

Dyes	Fibre Type	Loss in Effluent (%)	Degree of Fixation (%)
Acid	Polyamide	5–20	80–95
Disperse	Polyester	0–10	90–100
Basic	Acrylic	0–5	95–100
Direct	Cellulose	5–30	70–95
Reactive	Cellulose	10–50	50–90

**Table 2 polymers-13-03009-t002:** Removal efficiency of cationic and anionic dyes using various types of chitosan composites under different operating conditions.

Chitosan Composites	Cationic/Anionic Dyes	pH	Biosorbent Dosage (g/L)	Initial Dyes Concentration (mg/L)	Contact Time (min)	Temperature (°C)	Dye Removal Efficiency (%)	References
**Chitosan/Zeolite**	Cationic	9	2.0	100	138.65	30	84.85	Metin, Çiftçi and Alver [54]
**Chitosan/ZnO**	Cationic	8	2.4	50	180	30	98.50	Muinde et al. [76]
**Chitosan/Activated Carbon**	Cationic	11	1.0	400	60	30	91.02	Fatombi et al. [82]
**Chitosan/Bentonite**	Cationic	10	0.2	100	360	25	88.00	Dotto et al. [75]
**Chitosan/Fe_3_O_4_/** **Graphene Oxide**	Cationic	11	1.0	100	70	27	87.6	Tran et al. [18]
**Chitosan/MgO**	Anionic	6.8	9.3	100	120	30	79.50	Nga, Thuy Chau and Viet [64]
**Chitosan/PVA**	Anionic	6	2.0	50	40	30	86.70	Anitha, Kumar and Kumar [47]
**Chitosan/Cellulose**	Anionic	6.6	2.5	500	625	30	95.00	Wang et al. [80]
**Chitosan/Kaolin Clay**	Anionic	4	0.6	140	30	30	99.50	Xie et al. [53]
**Chitosan/Graphene Oxide**	Anionic	2	1.0	250	1440	25	86.00	Kamal et al. [89]

**Table 3 polymers-13-03009-t003:** Kinetic constants of PFO and PSO kinetic models for biosorption of various dyes onto chitosan composites.

Chitosan Composites	Type of Dyes	*q_e,exp_* (mg/g)	PFO Model			PSO Model			References
			*q_e,cal_* (mg/g)	*k_1_* (min^−1^)	R^2^	*q_e,cal_* (mg/g)	*k_2_* (g/(mg·min))	R^2^	
Chitosan/Quartzite	Reactive Black	20.15	18.91	6.100	0.9682	21.55	0.0003	0.9918	Coura, Profeti and Profeti [78]
Chitosan/Zeolite	Methylene blue	23.04	7.40	0.0820	0.8000	23.75	0.0210	0.9970	Dehghani et al. [37]
Chitosan/Activated Carbon	Indigo carmine	23.98	7.80	0.7930	0.7790	2.04	0.8650	0.9980	Fatombi et al. [82]
Chitosan/Cellulose	Congo red	381.70	376.70	0.0106	0.8552	419.20	5.04 × 10^−6^	0.9529	Wang et al. [80]
Chitosan/ Graphene Oxide	Congo red	2.53	2.53	0.2440	0.8680	2.35	0.8650	0.9990	Kamal et al. [91]
Chitosan/MgO	Reactive blue	10.47	6.07	0.0051	0.7287	8.55	0.0045	0.9775	Nga, Thuy Chau and Viet [64]
Chitosan/ZnO	Acid Black 26	92.00	53.46	0.4190	0.9360	100.00	0.0330	1.0000	Salehi et al. [100]
Chitosan/Clay	Rose FRN	12.12	12.01	0.1770	0.9140	12.26	0.0358	0.9560	Kausar et al. [74]
Chitosan/Kaolin-Clay	Reactive blue	312.40	296.10	0.0640	0.9200	316.60	0.0320	0.9800	Jawad et al. [83]
Chitosan/MWCNT	Direct blue	30.12	3.90	0.0083	0.9670	30.12	0.0007	0.9730	Abbasi and Habibi [79]
Chitosan/Alumina	Methyl orange	41.76	10.14	0.0490	0.8430	43.59	0.0070	0.9990	Zhang, Zhou and Ou [8]
Chitosan/Bentonite	Amido Black 10B	239.90	121.10	0.0093	0.9840	246.90	0.0002	0.9965	Liu et al. [101]

**Table 4 polymers-13-03009-t004:** Intraparticle diffusion model parameters for biosorption of dyes by chitosan composites.

Chitosan Composites	Types of Dyes	Intraparticle Diffusion Model		References
		*k_p_* (mg/(g·min^0.5^))	*C* (mg/g)	
Chitosan/Zeolite	Methylene blue	0.501	19.23	Dehghani et al. [68]
Chitosan/Activated Carbon	Indigo carmine	1.648	20.865	Fatombi et al. [82]
Chitosan/Cellulose	Congo red	0.1553	389.9	Wang et al. [80]
Chitosan/ZnO	Acid black	12.84	38.9	Salehi et al. [100]
Chitosan/PVA	Methyl orange	1.517	9.343	Habiba et al. [103]
Chitosan/ γ-Fe_2_O_3_	Methyl orange	0.63362	14.61	Zhu et al. [67]

**Table 5 polymers-13-03009-t005:** Adsorption isotherm parameters of Langmuir and Freundlich isotherm model for biosorption of various dyes by different types of chitosan composites.

Chitosan Composites	Type of Dyes	*q_max_* (mg/g)	Langmuir Isotherm Model			Freundlich Isotherm Model			References
			*K_L_* (L/mg)	*R_L_*	R^2^	*K_F_* (mg/g)(L/g)^1/n^	*n*	R^2^	
Chitosan/Quartzite	Reactive Black	41.67	0.0300	0.4762	0.9580	9.17	3.900	0.9632	Dehghani et al. [37]
Chitosan/Zeolite	Methylene blue	24.51	0.3030	0.1160	0.9560	8.82	3.788	0.9990	Fatombi et al. [82]
Chitosan/Activated Carbon	Indigo carmine	208.33	0.0230	0.4600	0.9954	14.18	1.190	0.9878	Wang et al. [80]
Chitosan/Cellulose	Congo red	381.70	0.2633	0.0047	0.9729	132.90	5.488	0.8158	Kamal et al. [91]
Chitosan/Graphene Oxide	Congo red	370.37	0.0366	0.0546	0.9820	10.84	1.818	0.9160	Coura, Profeti and Profeti [78]
Chitosan/ZnO	Acid black	227.30	0.0482	0.2075	0.9960	101.13	7.905	0.9620	Nguyen, Nguyen and Nguyen [105]
Chitosan/MgO	Reactive blue	408.16	0.0127	0.4400	0.9544	5.55	1.090	0.9992	Nga, Thuy Chau and Viet [64]
Chitosan/PVA	Methyl orange	52.10	0.0846	0.1912	0.9998	4.40	1.578	0.9946	Habiba et al. [103]
Chitosan/Kaolin Clay	Reactive blue	560.90	0.0060	0.4167	0.9600	11.80	1.600	0.9400	Jawad et al. [83]
Chitosan/MWCNT	Direct blue	29.33	0.3860	0.3860	0.9980	10.42	3.650	0.9660	Abbasi and Habibi [79]
Chitosan/Alumina	Methyl orange	32.67	0.8210	0.8210	0.9880	11.27	2.884	0.9720	Zhang, Zhou and Ou [8]
Chitosan/Bentonite	Methylene blue	496.40	0.1660	0.0197	0.9985	170.40	5.156	0.9251	Liu et al. [101]

**Table 6 polymers-13-03009-t006:** Thermodynamic parameters for biosorption of dyes using various type of chitosan composites.

Chitosan Composites	Type of Dyes	Equilibrium Constant, *K_c_*	Gibbs’ Free Energy Change $\Delta G^{{}^{\circ}}$ (kJ/mol)	Enthalpy Change $\Delta H^{{}^{\circ}}$ (kJ/mol)	Entropy Change $\Delta S^{{}^{\circ}}$ (J/K/mol)	Temperature, *T* (K)	References
Chitosan/Quartzite	Reactive black	4.943 × 10^4^	−25.88	21.76	166.50	288	Dehghani et al. [37]
Chitosan/Activated Carbon	Indigo carmine	1.090 × 10^4^	−23.42	10.66	112.40	303	Wang et al. [80]
Chitosan/Cellulose	Congo red	7.899	−5.209	86.25	300.90	303	Kamal et al. [91]
Chitosan/MgO	Reactive blue	7.956 × 10^3^	−21.73	14.56	0.1250	291	Nga, Thuy Chau and Viet [64]
Chitosan/PVA	Eosin yellow	7.743	−5.156	−7.486	−7.716	303	Habiba et al. [103]
Chitosan/Zeolite	Methyl orange	2.730	−2.530	7.179	34.3048	283	Hussain et al. [109]
Crosslinked Chitosan Epichlorohydrin	Reactive red	2.593	−2.400	27.7	0.0990	303	Jawad et al. [108]
Chitosan/Kaolin Clay	Reactive blue	6.063	−4.540	4.810	0.4300	303	Jawad et al. [83]
Chitosan/Alumina	Methyl orange	5.335	−4.148	−10.54	−21.59	298	Zhang, Zhou and Ou [8]
Chitosan/Bentonite	Amido black	1.116	−0.2670	15.78	0.0547	293	Liu et al. [101]
Chitosan/γ-Fe_2_O_3_	Methyl orange	1.939	−1.762	−17.41	−48.89	320	Zhu et al. [67]

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
