# Peer review of "A State-of-the-Art Review on Biowaste Derived Chitosan Biomaterials for Biosorption of Organic Dyes: Parameter Studies, Kinetics, Isotherms and Thermodynamics"

_polymers, 2021, doi:10.3390/polym13173009_

Round 1
Reviewer 1 Report
Concerning the review article polymers-1356857, it is well written, including extensive information about the application of chitosan for biosorption of organic dyes. However, a few comments appended below should be considered prior to publishing in the Polymers Journal:
- The authors should update the references, particularly those recently published in 2021. You can find a lot of relevant articles in the polymer journal.
- The authors should highlight the extraction procedures of chitin from wastes and the further approaches for chitosan production. Moreover, please explain whether the molecular weights of chitosan affect its utilization as a biosorbent material.
- Fig. 1 should be improved.
- The authors are encouraged to reproduce and incorporate some figures from previously published articles with permission from publishers whether it is possible.
Author Response
Concerning the review article polymers-1356857, it is well written, including extensive information about the application of chitosan for biosorption of organic dyes. However, a few comments appended below should be considered prior to publishing in the Polymers Journal:
1. The authors should update the references, particularly those recently published in 2021. You can find a lot of relevant articles in the polymer journal.
Response: As requested by the reviewer, several relevant articles published in the polymers journal had been cited in this manuscript.
Marotta, A.; Luzzi, E.; de Luna, M.S.; Aprea, P.; Ambrogi, V.; Filippone, G. Chitosan/Zeolite Composite Aerogels for a Fast and Effective Removal of Both Anionic and Cationic Dyes from Water, Polym. 2021, 13, 1691.
Ramakrishnan, R.K.; Padil, V.V.T.; WacÅ‚awek, S.; ÄŒerník, M.; Varma, R.S. Eco-Friendly and Economic, Adsorptive Removal of Cationic and Anionic Dyes by Bio-Based Karaya Gum—Chitosan Sponge, Polym. 2021, 13(2), 251.
Sivanesan, I.; Gopal, J.; Muthu, M.; Shin, J.; Mari, S.; Oh, J.; Green Synthesized Chitosan/Chitosan Nanoforms/Nanocomposites for Drug Delivery Applications, Polym. 2021, 13, 2256.
El-Ghoul, Y.; Ammar, C.; Alminderej, F.M.; Shafiquzzaman, M. Design and Evaluation of a New Natural Multi-Layered Biopolymeric Adsorbent System-Based Chitosan/Cellulosic Nonwoven Material for the Biosorption of Industrial Textile Effluents, Polym. 2021, 13, 322.
Thamer, B.M,; Aldalbahi, A.; Moydeen M.A.; El-Newehy, M.H. In Situ Preparation of Novel Porous Nanocomposite Hydrogel as Effective Adsorbent for the Removal of Cationic Dyes from Polluted Water, Polym. 2020, 12, 3002.
Criado-Gonzalez, M.; Mijangos, C.; Hernández, R. Polyelectrolyte Multilayer Films Based on Natural Polymers: From Fundamentals to Bio-Applications, Polym. 2021, 13, 2254.
2. The authors should highlight the extraction procedures of chitin from wastes and the further approaches for chitosan production. Moreover, please explain whether the molecular weights of chitosan affect its utilization as a biosorbent material.
Response: According to the literature, the isolation of chitosan is commonly carried out mainly using chemical and biological methods. The chemical methods usually involve strong acids and bases to dissolve calcium carbonates and proteins, respectively. Meanwhile, biological treatments offer an attractive alternative way to extract chitin and chitosan due to its elimination of acidic and alkali treatments but encountered a longer extraction time as compared to chemical methods. Both methods involve three major steps as demineralization, deproteinization and deacetylation. On the other hand, decolouration process can be added to eliminate pigments such as Astaxanthin and β-carotene.
No extraction of chitin and chitosan production and characterization testing were studied in this review paper as the scope was only to review the application of food-waste derived chitosan for composite materials preparation together with their applications in environmental wastewater rather than the individual extraction methods. Another reason is such chitin and chitosan extraction review studies had been published previously in POLYMERS, MDPI [1] and other publisher i.e. Elsevier [2, 3].
In general, chitosan has molecular weight typically ranges from 20 to 1200 kDa and can be classified as low (<50 kDa), medium (50 to 250 kDa), and high (>250 kDa) molecular weight. The molecular weight of chitosan plays an important role on nanofiber development as it affects directly the chains entanglement [4]. Lee and Schlautman [5] reported that the longer and higher molecular weight of polymeric chain structures, increased the adsorption capacity and flocculation capability. However, the preparation and handling of working solutions for excessive high molecular weight of polymer became difficult due to the development of polymeric chain entanglements, which might avoid the formation of nanofiber and decrease the adsorption capacity.
Even though high molecular weight chitosan produces a more thermally and chemically stable particle compared to low molecular weight chitosan, it encountered increased aggregation of particles [6]. Moreover, chitosan with high molecular weight forms a rigid crystalline structure and increase the chitosan crystallinity, owing to the presence of huge inter and intramolecular hydrogen bonds along with its structure [7]. The crystallinity would control the polymer hydration, which in turn will determine the accessibility to active sites, lower the solubility and restrict its application as effective adsorbent. Since last few years, researchers have attracted the attention toward the preparation of low molecular weight of chitosan, to improve the solubility of chitosan and to expand its potential applications in more extensive fields [8].
Such related information had been included in section 1. Introduction and section 9. Future Prospects and Conclusion.
3. Fig. 1 should be improved.
Response: As requested by reviewer, Fig. 1 had been improved.
4. The authors are encouraged to reproduce and incorporate some figures from previously published articles with permission from publishers whether it is possible.
Response: As requested by reviewer, all the related figures from previously published articles are reproduced by obtaining permission from the publisher.
Reviewer 2 Report
The quality and typesetting of the figures in the paper could be improved to make a better review.
Author Response
The quality and typesetting of the figures in the paper could be improved to make a better review.
Response: As requested by the reviewer, the quality and typesetting of the figures in the paper had been improved to make a better review. Thank you for the comment.
Reviewer 3 Report
Title of the manuscript:
A state-of-the-art review on biowaste derived chitosan bio-materials for biosorption of organic dyes: Parameter studies, kinetics, isotherms and thermodynamics
Comments to the authors:
The topic is very interesting and well presented. The work clearly presents the review of the results by numerous research on the removal of organic dyes through biosorption processes.
Please find my comments below. Some minor corrections should be made.
1) Introduction
It provides sufficient background and includes all relevant references. The references are cited correctly.
2) Biopolymers
Line 162 (page 5): Figure 2-chemical structure should be corrected to a) for chitin in and b) for cellulose. Now they are opposite, and it is not correct.
Line 186 (page 5): correct the names of the Figures for the chemical structures of chitin and chitosan. (Linked to the comment above).
4) Biosorption parameter studies
Line 568 (page 14): Table 2: correct the text in the 1st row, that the words will be in one line. Now they are divided: Bioroben t etc. The same is at the 1st column, where are mentioned chitosan components.
6) Bisorption kinetics
Line 682 (page 17): Table 3: the same as mentioned at the Table 2 comment above. Words, brackets, should be corrected.
4) Future perspectives and conclusion
The conclusion and future perspectives are well presented. All important findings are included in this chapter.
5) References:
Are cited correctly.
Author Response
Title of the manuscript:
A state-of-the-art review on biowaste derived chitosan bio-materials for biosorption of organic dyes: Parameter studies, kinetics, isotherms and thermodynamics
Comments to the authors:
The topic is very interesting and well presented. The work clearly presents the review of the results by numerous research on the removal of organic dyes through biosorption processes.
Please find my comments below. Some minor corrections should be made.
1) Introduction
It provides sufficient background and includes all relevant references. The references are cited correctly.
Response: Thank you for the kind review and comment.
2) Biopolymers
Line 162 (page 5): Figure 2-chemical structure should be corrected to a) for chitin in and b) for cellulose. Now they are opposite, and it is not correct.
Response: Thank you for spotting out the error. We are sorry for the careless mistake. The Figure 2 chemical structures had been revised accordingly.
Line 186 (page 5): correct the names of the Figures for the chemical structures of chitin and chitosan. (Linked to the comment above).
Response: The correction had been made accordingly. Thank you for the comment.
4) Biosorption parameter studies
Line 568 (page 14): Table 2: correct the text in the 1st row, that the words will be in one line. Now they are divided: Bioroben t etc. The same is at the 1st column, where are mentioned chitosan components.
Response: Thank you for the comment. As requested by the reviewer, the text in the 1st row and 1st column in Table 2 had been corrected so that the words were appeared in one line.
6) Biosorption kinetics
Line 682 (page 17): Table 3: the same as mentioned at the Table 2 comment above. Words, brackets, should be corrected.
Response: Thank you for the comment. As requested by the reviewer, the text in the 1st row and 1st column in Table 3 had been corrected so that the words were appeared in one line.
4) Future perspectives and conclusion
The conclusion and future perspectives are well presented. All important findings are included in this chapter.
Response: Thank you for the kind review and valuable comments.
5) References:
Are cited correctly.
Response: Thank you for the kind review and valuable comments.